# Near Minimax Optimal Players for the Finite-Time $3$-Expert Prediction Problem

**Yasin Abbasi-Yadkori**
Adobe Research

**Peter L. Bartlett**
UC Berkeley

**Victor Gabillon**
Queensland University of Technology

## Abstract

We study minimax strategies for the online prediction problem with expert advice. It has been conjectured that a simple adversary strategy, called COMB, is near optimal in this game for any number of experts. Our results and new insights make progress in this direction by showing that, up to a small additive term, COMB is minimax optimal in the finite-time three expert problem. In addition, we provide for this setting a new near minimax optimal COMB-based learner. Prior to this work, in this problem, learners obtaining the optimal multiplicative constant in their regret rate were known only when $K = 2$ or $K \to \infty$. We characterize, when $K = 3$, the regret of the game scaling as $\sqrt{8/(9\pi)T} \pm \log(T)^2$ which gives for the first time the optimal constant in the leading ($\sqrt{T}$) term of the regret.

## 1 Introduction

This paper studies the online prediction problem with expert advice. This is a fundamental problem of machine learning that has been studied for decades, going back at least to the work of Hannan [12] (see [4] for a survey). As it studies prediction under adversarial data the designed algorithms are known to be robust and are commonly used as building blocks of more complicated machine learning algorithms with numerous applications. Thus, elucidating the yet unknown optimal strategies has the potential to significantly improve the performance of these higher level algorithms, in addition to providing insight into a classic prediction problem. The problem is a repeated two-player zero-sum game between an adversary and a learner. At each of the $T$ rounds, the adversary decides the quality/gain of $K$ experts' advice, while simultaneously the learner decides to follow the advice of one of the experts. The objective of the adversary is to maximize the regret of the learner, defined as the difference between the total gain of the learner and the total gain of the best fixed expert.

**Open Problems and our Main Results.** Previously this game has been solved asymptotically as both $T$ and $K$ tend to $\infty$: asymptotically the upper bound on the performance of the state-of-the-art Multiplicative Weights Algorithm (MWA) for the learner matches the optimal multiplicative constant of the asymptotic minimax optimal regret rate $\sqrt{(T/2) \log K}$ [3]. However, for finite $K$, this asymptotic quantity actually overestimates the finite-time value of the game. Moreover, Gravin et al. [10] proved a matching lower bound $\sqrt{(T/2) \log K}$ on the regret of the classic version of MWA, additionally showing that the optimal learner does not belong an extended MWA family. Already, Cover [5] proved that the value of the game is of order of $\sqrt{T/(2\pi)}$ when $K = 2$, meaning that the regret of a MWA learner is $47\%$ larger that the optimal learner in this case. Therefore the question of optimality remains open for non-asymptotic $K$ which are the typical cases in applications.

In studying a related setting with $K = 3$, where $T$ is sampled from a geometric distribution with parameter $\delta$, Gravin et al. [9] conjectured that, for any $K$, a simple adversary strategy, called the COMB adversary, is asymptotically optimal ($T \to \infty$, or when $\delta \to 0$), and also excessively competitive for finite-time fixed $T$. The **COMB strategy** sorts the experts based on their cumulative gains and, with probability one half, assigns gain one to each expert in an odd position and gain zero

to each expert in an even position. With probability one half, the zeros and ones are swapped. The simplicity and elegance of this strategy, combined with its almost optimal performance makes it very appealing and calls for a more extensive study of its properties.

Our results and new insights make progress in this direction by showing that, for any fixed $T$ and up to small additive terms, COMB is minimax optimal in the finite-time three expert problem. Additionally and with similar guarantees, we provide for this setting a new near minimax optimal COMB-based learner. For $K = 3$, the regret of a MWA learner is $39\%$ larger than our new optimal learner[1]. In this paper we also characterize, when $K = 3$, the regret of the game as $\sqrt{8/(9\pi)T} \pm \log(T)^2$ which gives for the first time the optimal constant in the leading ($\sqrt{T}$) term of the regret. Note that the state-of-the-art non-asymptotic lower bound in [15] on the value of this problem is non informative as the lower bound for the case of $K = 3$ is a negative quantity.

**Related Works and Challenges.** For the case of $K = 3$, Gravin et al. [9] proved the exact minimax optimality of a COMB-related adversary in the geometrical setting, i.e. where $T$ is not fixed in advance but rather sampled from a geometric distribution with parameter $\delta$. However the connection between the geometrical setting and the original finite-time setting is not well understood, even asymptotically (possibly due to the large variance of geometric distributions with small $\delta$). Addressing this issue, in Section 7 of [8], Gravin et al. formulate the "Finite vs Geometric Regret" conjecture which states that the value of the game in the geometrical setting, $V_\alpha$, and the value of the game in the finite-time setting, $V_T$, verify $V_T = \frac{2}{\sqrt{\pi}} V_{\alpha=1/T}$. We resolve here the conjecture for $K = 3$.

Analyzing the finite-time expert problem raises new challenges compared to the geometric setting. In the geometric setting, at any time (round) $t$ of the game, the expected number of remaining rounds before the end of the game is constant (does not depend on the current time $t$). This simplifies the problem to the point that, when $K = 3$, there exists an exactly minimax optimal adversary that ignores the time $t$ and the parameter $\delta$. As noted in [9], and noticeable from solving exactly small instances of the game with a computer, in the finite-time case, the exact optimal adversary seems to depend in a complex manner on time and state. It is therefore natural to compromise for a simpler adversary that is optimal up to a small additive error term. Actually, based on the observation of the restricted computer-based solutions, the additive error term of COMB seems to vanish with larger $T$.

Tightly controlling the errors made by COMB is a new challenge with respect to [9], where the solution to the optimality equations led directly to the exact optimal adversary. The existence of such equations in the geometric setting crucially relies on the fact that the value-to-go of a given policy in a given state does not depend on the current time $t$ (because geometric distributions are memoryless). To control the errors in the finite-time setting, our new approach solves the game by backward induction showing the *approximate greediness* of COMB with respect to itself (read Section 2.1 for an overview of our new proof techniques and their organization). We use a novel exchangeability property, new connections to random walks and a close relation that we develop between COMB and a TWIN-COMB strategy. Additional connections with new related optimal strategies and random walks are used to compute the value of the game (Theorem 2). We discuss in Section 6 how our new techniques have more potential to extend to an arbitrary number of arms, than those of [9].

Additionally, we show how the approximate greediness of COMB with respect to itself is key to proving that a learner based directly on the COMB adversary is itself quasi-minimax-optimal. This is the first work to extend to the approximate case, approaches used to designed exactly optimal players in related works. In [2] a probability matching learner is proven optimal under the assumption that the adversary is limited to a fixed cumulative loss for the best expert. In [14] and [1], the optimal learner relies on estimating the value-to-go of the game through rollouts of the optimal adversary's plays. The results in these papers were limited to games where the optimal adversary was only playing canonical unit vector while our result holds for general gain vectors. Note also that a probability matching learner is optimal in [9].

**Notation:** Let $[a : b] = \{a, a + 1, \ldots, b\}$ with $a, b \in \mathbb{N}$, $a \leq b$, and $[a] = [1 : a]$. For a vector $\boldsymbol{w} \in \mathbb{R}^n$, $n \in \mathbb{N}$, $\|\boldsymbol{w}\|_\infty = \max_{k \in [n]} |\boldsymbol{w}_k|$. A vector indexed by both a time $t$ and a specific element index $k$ is $\boldsymbol{w}_{t,k}$. An undiscounted Markov Decision Process (MDP) [13, 16] $\mathcal{M}$ is a 4-tuple $\langle \mathcal{S}, \mathcal{A}, r, p \rangle$. $\mathcal{S}$ is the state space, $\mathcal{A}$ is the set of actions, $r : \mathcal{S} \times \mathcal{A} \to \mathbb{R}$ is the reward function, and the transition model $p(\cdot | s, a)$ gives the probability distribution over the next state when action $a$ is taken in state $s$. A state is denoted by $\boldsymbol{s}$ or $\boldsymbol{s}_t$ if it is taken at time $t$. An action is denoted by $a$ or $a_t$.

## 2 The Game

We consider a game, composed of $T$ rounds, between two players, called a learner and an adversary. At each time/round $t$ the learner chooses an index $I_t \in [K]$ from a distribution $p_t$ on the $K$ arms. Simultaneously, the adversary assigns a binary gain to each of the arms/experts, possibly at random from a distribution $\dot{A}_t$, and we denote the vector of these gains by $g_t \in \{0, 1\}^K$. The adversary and the learner then observe $I_t$ and $g_t$. For simplicity we use the notation $g_{[t]} = (g_s)_{s=1,\ldots,t}$. The value of one realization of such a game is the cumulative regret defined as

$$R_T = \left\| \sum_{t=1}^{T} g_t \right\|_{\infty} - \sum_{t=1}^{T} g_{t,I_t} \ .$$

A state $s \in \mathcal{S} = (\mathbb{N} \cup \{0\})^K$ is a $K$-dimensional vector such that the $k$-th element is the cumulative sum of gains dealt by the adversary on arm $k$ before the current time $t$. Here the state does not include $t$ but is typically denoted for a specific time $t$ as $s_t$ and computed as $s_t = \sum_{t'=1}^{t-1} g_{t'}$. This definition is motivated by the fact that there exist minimax strategies for both players that rely solely on the state and time information as opposed to the complete history of plays, $g_{[t]} \cup I_{[t]}$. In state $s$, the set of *leading experts*, i.e., those with maximum cumulative gain, is $\mathbf{X}(s) = \{k \in [K] : s_k = \|s\|_{\infty}\}$.

We use $\pi$ to denote the (possibly non-stationary) strategy/policy used by the adversary, i.e., for any input state $s$ and time $t$ it outputs the gain distribution $\pi(s, t)$ played by the adversary at time $t$ in state $s$. Similarly we use $\bar{p}$ to denote the strategy of the learner. As the state depends only on the adversary plays, we can sample a state $s$ at time $t$ from $\pi$.

Given an adversary $\pi$ and a learner $\bar{p}$, the expected regret of the game, $V_{\bar{p},\pi}^T$, is $V_{\bar{p},\pi}^T = \mathbb{E}_{g_{[T]} \sim \pi, I_{[T]} \sim \bar{p}} [R_T]$. The learner tries to minimize the expected regret while the adversary tries to maximize it. The value of the game is the minimax value $V_T$ defined by

$$V_T = \min_{\bar{p}} \max_{\pi} V_{\bar{p},\pi}^T = \max_{\pi} \min_{\bar{p}} V_{\bar{p},\pi}^T.$$

In this work, we are interested in the search for optimal minimax strategies, which are adversary strategies $\pi^*$ such that $V_T = \min_{\bar{p}} V_{\bar{p},\pi^*}^T$ and learner strategies $\bar{p}^*$, such that $V_T = \max_{\pi} V_{\bar{p}^*,\pi}^T$.

### 2.1 Summary of our Approach to Obtain the Near Greediness of COMB

Most of our material is new. First, Section 3 recalls that Gravin et al. [9] have shown that the search for the optimal adversary $\pi^*$ can be restricted to the finite family of *balanced strategies* (defined in the next section). When $K = 3$, the action space of a balanced adversary is limited to seven stochastic actions (gain distributions), denoted by $\dot{B}_3 = \{\dot{w}, \dot{c}, \dot{v}, \dot{1}, \dot{2}, \{\}, \{123\}\}$ (see Section 5.1 for their description). The COMB adversary repeats the gain distribution $\dot{c}$ at each time and in any state.

In Section 4 we provide an explicit formulation of the problem as finding $\pi^*$ inside an MDP with a specific reward function. Interestingly, we observe that another adversary, which we call TWIN-COMB and denote by $\pi_W$, which repeats the distribution $\dot{w}$, has the same value as $\pi_C$ (Section 5.1). To control the errors made by COMB, the proof uses a novel and intriguing exchangeability property (Section 5.2). This exchangeability property holds thanks to the surprising role played by the TWIN-COMB strategy. For any distributions $\dot{A} \in \dot{B}_3$ there exists a distribution $\dot{D}$, mixture of $\dot{c}$ and $\dot{w}$, such that for almost all states, playing $\dot{A}$ and then $\dot{D}$ is the same as playing $\dot{w}$ and then $\dot{A}$ in terms of the expected reward and the probabilities over the next states after these two steps. Using Bellman operators, this can be concisely written as: for any (value) function $f : \mathcal{S} \longrightarrow \mathbb{R}$, in (almost) any state $s$, we have that $[T_{\dot{A}}[T_{\dot{D}}f]](s) = [T_{\dot{w}}[T_{\dot{A}}f]](s)$. We solve the MDP with a backward induction in time from $t = T$. We show that playing $\dot{c}$ at time $t$ is almost greedy with respect to playing $\pi_C$ in later rounds $t' > t$. The greedy error is defined as the difference of expected reward between always playing $\pi_C$ and playing the best (greedy) first action before playing COMB. Bounding how these errors accumulate through the rounds relates the value of COMB to the value of $\pi^*$ (Lemma 16).

To illustrate the main ideas, let us first make two simplifying (but unrealistic) assumptions at time $t$: COMB has been proven greedy w.r.t. itself in rounds $t' > t$ and the exchangeability holds in all states. Then we would argue at time $t$ that by the exchangeability property, instead of optimizing the greedy

action w.r.t. COMB as $\max_{\dot{\mathrm{A}} \in \dot{\boldsymbol{B}}_3} \dot{\mathrm{A}}\dot{\mathrm{C}} \ldots \dot{\mathrm{C}}$, we can study the optimizer of $\max_{\dot{\mathrm{A}} \in \dot{\boldsymbol{B}}_3} \dot{\mathrm{W}}\dot{\mathrm{A}}\dot{\mathrm{C}} \ldots \dot{\mathrm{C}}$. Then we use the induction property to conclude that $\dot{\mathrm{C}}$ is the solution of the previous optimization problem.

Unfortunately, the exchangeability property does not hold in one specific state denoted by $\boldsymbol{s}_\alpha$. What saves us though is that we can directly compute the error of greedification of any gain distribution with respect to COMB in $\boldsymbol{s}_\alpha$ and show that it diminishes exponentially fast as $T - t$, the number of rounds remaining, increases (Lemma 7). This helps us to control how the errors accumulate during the induction. From one given state $\boldsymbol{s}_t \neq \boldsymbol{s}_\alpha$ at time $t$, first, we use the exchangeability property once when trying to assess the 'quality' of an action $\dot{\mathrm{A}}$ as a greedy action w.r.t. COMB. This leads us to consider the quality of playing $\dot{\mathrm{A}}$ in possibly several new states $\{\boldsymbol{s}_{t+1}\}$ at time $t+1$ reached following TWIN-COMB in $\boldsymbol{s}$. We use our exchangeability property repeatedly, starting from the state $\boldsymbol{s}_t$ until a subsequent state reaches $\boldsymbol{s}_\alpha$, say at time $t_\alpha$, where we can substitute the exponentially decreasing greedy error computed at this time $t_\alpha$ in $\boldsymbol{s}_\alpha$. Here the subsequent states are the states reached after having played TWIN-COMB repetitively starting from the state $\boldsymbol{s}_t$. If $\boldsymbol{s}_\alpha$ is never reached we use the fact that COMB is an optimal action everywhere else in the last round. The problem is then to determine at which time $t_\alpha$, starting from any state at time $t$ and following a TWIN-COMB strategy, we hit $\boldsymbol{s}_\alpha$ for the first time. This is translated into a classical *gambler's ruin* problem, which concerns the hitting times of a simple random walk (Section 5.3). Similarly the value of the game is computed using the study of the expected number of equalizations of a simple random walk (Theorem 5.1).

## 3 Solving for the Adversary Directly

In this section, we recall the results from [9] that, for arbitrary $K$, permit us to directly search for the minimax optimal adversary in the restricted set of *balanced* adversaries while ignoring the learner.

**Definition 1.** *A gain distribution $\dot{\mathrm{A}}$ is balanced if there exists a constant $c_{\dot{\mathrm{A}}}$, the mean gain of $\dot{\mathrm{A}}$, such that $\forall k \in [K]$, $c_{\dot{\mathrm{A}}} = \mathrm{E}_{\boldsymbol{g}|\dot{\mathrm{A}}}[\boldsymbol{g}_k]$. A balanced adversary uses exclusively balanced gain distributions.*

**Lemma 1** (Claim 5 in [9]). *There exists a minimax optimal balanced adversary.*

Use $\boldsymbol{B}$ to denote the set of all balanced strategies and $\dot{\boldsymbol{B}}$ to denote the set of all balanced gain distributions. Interestingly, as demonstrated in [9], a balanced adversary $\pi$ inflicts the same regret on every learner: If $\pi \in \boldsymbol{B}$, then $\exists V_T^\pi \in \mathbb{R} : \forall \bar{\boldsymbol{p}}, V_{\bar{\boldsymbol{p}},\pi}^T = V_T^\pi$. (See Lemma 10) Therefore, given an adversary strategy $\pi$, we can define the value-to-go $V_{t_0}^\pi(\boldsymbol{s})$ associated with $\pi$ from time $t_0$ in state $\boldsymbol{s}$,

$$V_{t_0}^\pi(\boldsymbol{s}) = \underset{\boldsymbol{s}_{T+1}}{\mathrm{E}} \|\boldsymbol{s}_{T+1}\|_\infty - \sum_{t=t_0}^{T} \underset{\boldsymbol{s}_t}{\mathrm{E}} \left[ c_{\pi(\boldsymbol{s}_t,t)} \right], \quad \boldsymbol{s}_{t+1} \sim P(.|\boldsymbol{s}_t, \pi(\boldsymbol{s}_t, t), \boldsymbol{s}_{t_0} = \boldsymbol{s}).$$

Another reduction comes from the fact that the set of balanced gain distributions can be seen as a convex combination of a finite set of balanced distributions [9, Claim 2 and 3]. We call this limited set the atomic gain distributions. Therefore the search for $\pi^*$ can be limited to this set. The set of convex combinations of the $m$ distributions $\dot{\mathrm{A}}_1, \ldots \dot{\mathrm{A}}_m$ is denoted by $\Delta(\dot{\mathrm{A}}_1, \ldots \dot{\mathrm{A}}_m)$.

## 4 Reformulation as a Markovian Decision Problem

In this section we formulate, for arbitrary $K$, the maximization problem over balanced adversaries as an undiscounted MDP problem $\langle \boldsymbol{\mathcal{S}}, \boldsymbol{\mathcal{A}}, r, p \rangle$. The state space $\boldsymbol{\mathcal{S}}$ was defined in Section 2 and the action space is the set of atomic balanced distributions as discussed in Section 3. The transition model is defined by $p(.|\boldsymbol{s}, \dot{\mathrm{D}})$, which is a probability distribution over states given the current state $\boldsymbol{s}$ and a balanced distribution over gains $\dot{\mathrm{D}}$. In this model, the transition dynamics are deterministic and entirely controlled by the adversary's action choices. However, the adversary is forced to choose stochastic actions (balanced gain distributions). The maximization problem can therefore also be thought of as designing a balanced random walk on states so as to maximize a sum of rewards (that are yet to be defined). First, we define $P_{\dot{\mathrm{A}}}$ the transition probability operator with respect to a gain distribution $\dot{\mathrm{A}}$. Given function $f : \boldsymbol{\mathcal{S}} \longrightarrow \mathbb{R}$, $P_{\dot{\mathrm{A}}}$ returns

$$[P_{\dot{\mathrm{A}}} f](\boldsymbol{s}) = \mathrm{E}[f(\boldsymbol{s}')|\boldsymbol{s}' \sim p(.|\boldsymbol{s}, \dot{\mathrm{A}})] = \underset{\boldsymbol{g} \sim \boldsymbol{s}, \dot{\mathrm{A}}}{\mathrm{E}}[f(\boldsymbol{s} + \boldsymbol{g})].$$

$\boldsymbol{g}$ is sampled in $\boldsymbol{s}$ according to $\dot{\mathrm{A}}$. Given $\dot{\mathrm{A}}$ in $\boldsymbol{s}$, the per-step regret is denoted by $r_{\dot{\mathrm{A}}}(\boldsymbol{s})$ and defined as

$$r_{\dot{\mathrm{A}}}(\boldsymbol{s}) = \underset{\boldsymbol{s}'|\boldsymbol{s}, \dot{\mathrm{A}}}{\mathrm{E}} \|\boldsymbol{s}'\|_\infty - \|\boldsymbol{s}\|_\infty - c_{\dot{\mathrm{A}}}.$$

Given an adversary strategy $\pi$, starting in $\boldsymbol{s}$ at time $t_0$, the cumulative per-step regret is $\bar{V}_{t_0}^\pi(\boldsymbol{s}) = \sum_{t=t_0}^T \mathrm{E}\left[r_{\pi(\cdot,t)}(\boldsymbol{s}_t) \mid \boldsymbol{s}_{t+1} \sim p(.|\boldsymbol{s}_t, \pi(\boldsymbol{s}_t, t)), \boldsymbol{s}_{t_0} = \boldsymbol{s})\right]$. The action-value function of $\pi$ at $(s, \dot{\mathrm{D}})$ and $t$ is the expected sum of rewards received by starting from $\boldsymbol{s}$, taking action $\dot{\mathrm{D}}$, and then following $\pi$: $\bar{Q}_t^\pi(\boldsymbol{s}_t, \dot{\mathrm{D}}) = \mathrm{E}\left[\sum_{t'=t}^T r_{\dot{\mathrm{A}}_t}(\boldsymbol{s}_t) \mid \dot{\mathrm{A}}_0 = \dot{\mathrm{D}}, s_{t+1} \sim p(\cdot|\boldsymbol{s}_t, \dot{\mathrm{A}}_t), \dot{\mathrm{A}}_{t+1} = \pi(\boldsymbol{s}_{t+1}, t+1)\right]$. The Bellman operator of $\dot{\mathrm{A}}$, $T_{\dot{\mathrm{A}}}$, is $[T_{\dot{\mathrm{A}}} f](\boldsymbol{s}) = r_{\dot{\mathrm{A}}}(\boldsymbol{s}) + [P_{\dot{\mathrm{A}}} f](\boldsymbol{s})$. with $[T_{\pi(\boldsymbol{s},t)} \bar{V}_{t+1}^\pi](\boldsymbol{s}) = \bar{V}_t^\pi(\boldsymbol{s})$.

This per-step regret, $r_{\dot{\mathrm{A}}}(\boldsymbol{s})$, depends on $\boldsymbol{s}$ and $\dot{\mathrm{A}}$ and not on the time step $t$. Removing the time from the picture permits a simplified view of the problem that leads to a natural formulation of the exchangeability property that is independent of the time $t$. Crucially, this decomposition of the regret into per-step regrets is such that maximizing $\bar{V}_{t_0}^\pi(\boldsymbol{s})$ over adversaries $\pi$ is equivalent, for all time $t_0$ and $\boldsymbol{s}$, to maximizing over adversaries the original value of the game, the regret $V_{t_0}^\pi(\boldsymbol{s})$ (Lemma 2).

**Lemma 2.** *For any adversary strategy $\pi$ and any state $\boldsymbol{s}$ and time $t_0$, $V_{t_0}^\pi(\boldsymbol{s}) = \bar{V}_{t_0}^\pi(\boldsymbol{s}) + \|\boldsymbol{s}\|_\infty$.*

The proof of Lemma 2 is in Section 8. In the following, our focus will be on maximizing $\bar{V}_t^\pi(\boldsymbol{s})$ in any state $\boldsymbol{s}$. We now show some basic properties of the per-step regret that holds for an arbitrary number of experts $K$ and discuss their implications. The proofs are in Section 9.

**Lemma 3.** *Let $\dot{\mathrm{A}} \in \dot{\boldsymbol{B}}$, for all $\boldsymbol{s}, t$, we have $0 \le r_{\dot{\mathrm{A}}}(\boldsymbol{s}) \le 1$. Furthermore if $|\mathbf{X}(\boldsymbol{s})| = 1$, $r_{\dot{\mathrm{A}}}(\boldsymbol{s}) = 0$.*

Lemma 3 shows that a state $\boldsymbol{s}$ in which the reward is not zero contains at least two equal leading experts, $|\mathbf{X}(\boldsymbol{s})| > 1$. Therefore the goal of maximizing the reward can be rephrased into finding a policy that visits the states with $|\mathbf{X}(\boldsymbol{s})| > 1$ as often as possible, while still taking into account that the per-step reward increases with $|\mathbf{X}(\boldsymbol{s})|$. The set of states with $|\mathbf{X}(\boldsymbol{s})| > 1$ is called the 'reward wall'.

**Lemma 4.** *In any state $\boldsymbol{s}$, with $|\mathbf{X}(\boldsymbol{s})| = 2$, for any balanced gain distribution $\dot{\mathrm{D}}$ such that with probability one exactly one of the leading expert receives a gain of 1, $r_{\dot{\mathrm{D}}}(\boldsymbol{s}) = \max_{\dot{\mathrm{A}} \in \dot{\boldsymbol{B}}} r_{\dot{\mathrm{A}}}(\boldsymbol{s})$.*

# 5 The Case of $K = 3$

## 5.1 Notations in the 3-Experts Case, the COMB and the TWIN-COMB Adversaries

First we define the state space in the 3-expert case. The experts are sorted with respect to their cumulative gains and are named in decreasing order, the leading expert, the middle expert and the lagging expert. As mentioned in [9], in our search for the minimax optimal adversary, it is sufficient for any $K$ to describe our state only using $d_{ij}$ that denote the difference between the cumulative gains of consecutive sorted experts $i$ and $j = i + 1$. Here, $i$ denotes the expert with $i$th largest cumulative gains, and hence $d_{ij} \ge 0$ for all $i < j$. Therefore one notation for a state, that will be used throughout this section, is $\boldsymbol{s} = (x, y) = (d_{12}, d_{23})$. We distinguish four types of states $C_1, C_2, C_3, C_4$ as detailed below in Figure 1. In the same figure, in the center, the states are represented on a 2d-grid. $C_4$ contains only the state denoted $\boldsymbol{s}_\alpha = (0, 0)$.

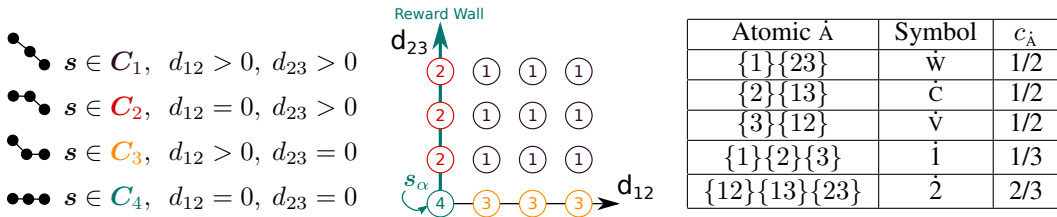

Figure 1: 4 types of states (left), their location on the 2d grid of states (center) and 5 atomic $\dot{\mathrm{A}}$ (right)

Concerning the action space, the gain distributions use brackets. The group of arms in the same bracket receive gains together and each group receive gains with equal probability. For instance, $\{1\}\{2\}\{3\}$ exclusively deals a gain to expert 1 (leading expert) with probability $1/3$, expert 2 (middle expert) with probability $1/3$, and expert 3 (lagging expert) with probability $1/3$, whereas $\{1\}\{23\}$ means dealing a gain to expert 1 alone with probability $1/2$ and experts 2 and 3 together with probability $1/2$. As discussed in Section 3, we are searching for a $\pi^*$ using mixtures of atomic balanced distributions. When $K = 3$ there are seven atomic distributions, denoted by $\dot{\boldsymbol{B}}_3 = \{\dot{\mathrm{v}}, \dot{1}, \dot{2}, \dot{\mathrm{c}}, \dot{\mathrm{w}}, \{\}, \{123\}\}$ and described in Figure 1 (right). Moreover, in Figure 2, we report in detail—in a table (left) and

| $s$ | $r_{\dot{\text{C}}}(s)$ | Distribution of next state $s' \sim p(\cdot|s, \dot{\text{C}})$ with $s = (x, y)$ |
|---|---|---|
| $C_1$ | 0 | $P(s' = (x-1, y+1)) = P(s' = (x+1, y-1)) = .5$ |
| $C_2$ | 1/2 | $P(s' = (x+1, y)) = P(s' = (x+1, y-1)) = .5$ |
| $C_3$ | 0 | $P(s' = (x, y+1)) = P(s' = (x-1, y+1)) = .5$ |
| $C_4$ | 1/2 | $P(s' = (x, y+1)) = P(s' = (x+1, y)) = .5$ |

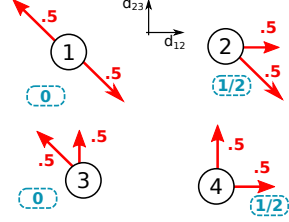

Figure 2: The per-step regret and transition probabilities of the gain distribution $\dot{\text{C}}$

an illustration (right) on the 2-D state grid—the properties of the COMB gain distribution $\dot{\text{C}}$. The remaining atomic distributions are similarly reported in the appendix in Figures 5 to 8.

In the case of three experts, the COMB distribution is simply playing {2}{13} in any state. We use $\dot{\text{W}}$ to denote the strategy that plays {1}{23} in any state and refer to it as the TWIN-COMB strategy. The COMB and TWIN-COMB *strategies* (as opposed to the distributions) repeat their respective gain distributions in any state and any time. They are respectively denoted $\pi_{\text{C}}, \pi_{\text{W}}$. The Lemma 5 shows that the COMB strategy $\pi_{\text{C}}$, the TWIN-COMB strategy $\pi_{\text{W}}$ and therefore any mixture of both, have the same expected cumulative per-step regret. The proof is reported to Section 11.

**Lemma 5.** *For all states $s$ at time $t$, we have $\bar{V}_t^{\pi_{\text{C}}}(s) = \bar{V}_t^{\pi_{\text{W}}}(s)$.*

### 5.2 The Exchangeability Property

**Lemma 6.** *Let $\dot{\text{A}} \in \dot{B}_3$, there exists $\dot{\text{D}} \in \Delta(\dot{\text{C}}, \dot{\text{W}})$ such that for any $s \neq s_\alpha$, and for any $f : \mathcal{S} \longrightarrow \mathbb{R}$,*

$$[T_{\dot{\text{A}}}[T_{\dot{\text{D}}}f]](s) = [T_{\dot{\text{W}}}[T_{\dot{\text{A}}}f]](s).$$

*Proof.* If $\dot{\text{A}} = \dot{\text{W}}$, $\dot{\text{A}} = \{\}$ or $\dot{\text{A}} = \{123\}$, use $\dot{\text{D}} = \dot{\text{W}}$. If $\dot{\text{A}} = \dot{\text{C}}$, use Lemma 11 and 12.
**Case 1. $\dot{\text{A}} = \dot{\text{V}}$:** $\dot{\text{V}}$ is equal to $\dot{\text{C}}$ in $C_3 \cup C_4$ and if $s' \sim p(.|s, \dot{\text{W}})$ with $s \in C_3$ then $s' \in C_3 \cup C_4$. So when $s \in C_3$ we reuse the case $\dot{\text{A}} = \dot{\text{C}}$ above. When $s \in C_1 \cup C_2$, we consider two cases.
**Case 1.1. $s \neq (0, 1)$:** We choose $\dot{\text{D}} = \dot{\text{W}}$ which is {1}{23}. If $s' \sim p(.|s, \dot{\text{V}})$ with $s \in C_2$ then $s' \in C_2$. Similarly, if $s' \sim p(.|s, \dot{\text{V}})$ with $s \in C_1$ then $s' \in C_1 \cup C_3$. Moreover $\dot{\text{D}}$ modifies similarly the coordinates $(d_{12}, d_{23})$ of $s \in C_1$ and $s \in C_3$. Therefore the effect in terms of transition probability and reward of $\dot{\text{D}}$ is the same whether it is done before or after the actions chosen by $\dot{\text{V}}$. If $s' \sim p(.|s, \dot{\text{D}})$ with $s \in C_1 \cup C_2$ then $s' \in C_1 \cup C_2$. Moreover $\dot{\text{V}}$ modifies similarly the coordinates $(d_{12}, d_{23})$ of $s \in C_1$ and $s \in C_2$. Therefore the effect in terms of the transition probability of $\dot{\text{V}}$ is the same whether it is done before or after the action $\dot{\text{D}}$. In terms of reward, notice that in the states $s \in C_1 \cup C_2$, $\dot{\text{V}}$ has 0 per-step regret and using $\dot{\text{V}}$ does not make $s'$ leave or enter the reward wall.
**Case 1.2 $s_t = (0, 1)$:** We can chose $\dot{\text{D}} = \dot{\text{W}}$. One can check from the tables in Figures 7 and 8 that exchangebility holds. Additionally we provide an illustration of the exchangeability equality in the 2d-grid in Figure 1. The starting state $s = (0, 1)$, is graphically represented by ■. We show on the grid the effect of the gain distribution $\dot{\text{V}}$ (in dashed red) followed (left picture) or preceded (right picture) by the gain distribution $\dot{\text{D}}$ (in plain blue). The illustration shows that $\dot{\text{V}}{\cdot}\dot{\text{D}}$ and $\dot{\text{D}}{\cdot}\dot{\text{V}}$ lead to the same final states (○○) with equal probabilities. The rewards are displayed on top of the pictures. Their color corresponds to the actions, the probabilities are in italic, and the rewards are in roman.

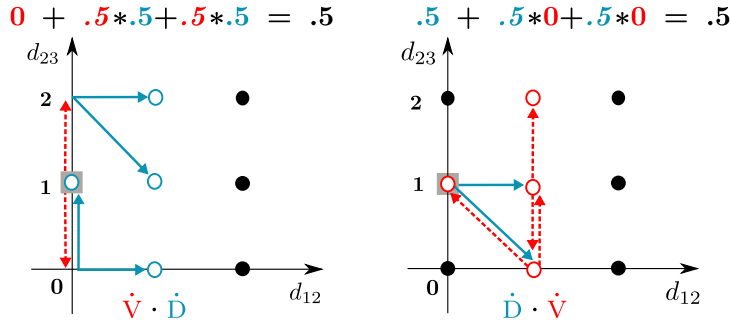

**Case 2 & 3. $\dot{\text{A}} = \dot{1}$ & $\dot{\text{A}} = \dot{2}$:** The proof is similar and is reported in Section 12 of the appendix. □

## 5.3 Approximate Greediness of COMB, Minimax Players and Regret

The greedy error of the gain distribution $\dot{\text{D}}$ in state $\boldsymbol{s}$ at time $t$ is

$$\epsilon^{\dot{\text{D}}}_{\boldsymbol{s},t} \;=\; \max_{\dot{\text{A}}\in\dot{\boldsymbol{B}}_3} \bar{Q}^{\pi_{\text{C}}}_t(\boldsymbol{s},\dot{\text{A}}) - \bar{Q}^{\pi_{\text{C}}}_t(\boldsymbol{s},\dot{\text{D}}).$$

Let $\epsilon^{\dot{\text{D}}}_t \;=\; \max_{\boldsymbol{s}\in\mathcal{S}} \epsilon^{\dot{\text{D}}}_{\boldsymbol{s},t}$ denote the maximum greedy error of the gain distribution $\dot{\text{D}}$ at time $t$. The COMB greedy error in $\boldsymbol{s}_\alpha$ is controlled by the following lemma proved in Section 13.1. Missing proofs from this section are in the appendix in Section 13.2.

**Lemma 7.** *For any $t \in [T]$ and gain distribution $\dot{\text{D}} \in \{\dot{\text{W}}, \dot{\text{C}}, \dot{\text{V}}, \dot{\text{I}}\}$, $\epsilon^{\dot{\text{D}}}_{\boldsymbol{s}_\alpha,t} \leq \frac{1}{6}\left(\frac{1}{2}\right)^{T-t}$.*

The following proposition shows how we can index the states in the 2d-grid as a one dimensional line over which the TWIN-COMB strategy behaves very similarly to a simple random walk. Figure 3 (top) illustrates this random walk on the 2d-grid and the indexing scheme (the yellow stickers).

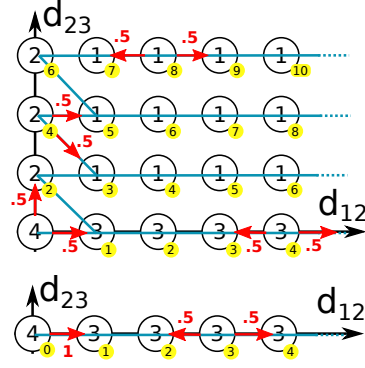

**Proposition 1.** *Index a state $\boldsymbol{s} = (x,y)$ by $i_{\boldsymbol{s}} = x + 2y$ irrespective of the time. Then for any state $\boldsymbol{s} \neq \boldsymbol{s}_\alpha$, and $\boldsymbol{s}' \sim p(\cdot|\boldsymbol{s},\dot{\text{W}})$ we have that $P(i_{\boldsymbol{s}'} = i_{\boldsymbol{s}}-1) = P(i_{\boldsymbol{s}'} = i_{\boldsymbol{s}}+1) = \frac{1}{2}$.*

Consider a random walk that starts from state $\boldsymbol{s}_0 = \boldsymbol{s}$ and is generated by the TWIN-COMB strategy, $\boldsymbol{s}_{t+1} \sim p(\cdot|\boldsymbol{s}_t,\dot{\text{W}})$. Define the random variable $T_{\alpha,\boldsymbol{s}} = \min\{t \in \mathbb{N}\cup\{0\} : \boldsymbol{s}_t = \boldsymbol{s}_\alpha\}$. This random variable is the number of steps of the random walk before hitting $\boldsymbol{s}_\alpha$ for the first time. Then, let $P_\alpha(\boldsymbol{s},t)$ be the probability that $\boldsymbol{s}_\alpha$ is reached after $t$ steps: $P_\alpha(\boldsymbol{s},t) = P(T_{\alpha,\boldsymbol{s}} = t)$. Lemma 8 controls the COMB greedy error in $\boldsymbol{s}_t$ in relation to $P_\alpha(\boldsymbol{s},t)$. Lemma 9 derives a state-independent upper-bound for $P_\alpha(\boldsymbol{s},t)$.

Figure 3: Numbering TWIN-COMB (top) & $\pi_{\text{G}}$ random walks (bottom)

**Lemma 8.** *For any time $t \in [T]$ and state $\boldsymbol{s}$,*

$$\epsilon^{\dot{\text{C}}}_{\boldsymbol{s},t} \;\leq\; \sum_{t'=t}^{T} P_\alpha(\boldsymbol{s},t'-t)\frac{1}{6}\left(\frac{1}{2}\right)^{T-t'}.$$

*Proof.* If $\boldsymbol{s} = \boldsymbol{s}_\alpha$, this is a direct application of Lemma 7 as $P_\alpha(\boldsymbol{s}_\alpha,t') = 0$ for $t' > 0$.

When $\boldsymbol{s} \neq \boldsymbol{s}_\alpha$, the following proof is by induction.

**Initialization:** Let $t = T$. At the last round only the last per-step regret matters (for all states $\boldsymbol{s}$, $\bar{Q}^{\pi_{\text{C}}}_t(\boldsymbol{s},\dot{\text{D}}) = r_{\dot{\text{D}}}(\boldsymbol{s})$). As $\boldsymbol{s} \neq \boldsymbol{s}_\alpha$, $\boldsymbol{s}$ is such that $|\mathbf{X}(\boldsymbol{s})|\leq 2$ then $r_{\dot{\text{D}}}(\boldsymbol{s}) = \max_{\dot{\text{A}}\in\dot{\boldsymbol{B}}} r_{\dot{\text{A}}}(\boldsymbol{s})$ because of Lemma 4 and Lemma 3. Therefore the statement holds.

**Induction:** Let $t < T$. We assume the statement is true at time $t+1$. We distinguish two cases.

For all gain distributions $\dot{\text{D}} \in \dot{\boldsymbol{B}}_3$,

$$\bar{Q}^{\pi_{\text{C}}}_t(\boldsymbol{s},\dot{\text{D}}) \overset{(a)}{=} [T_{\dot{\text{D}}}[T_{\dot{\text{E}}}\bar{V}^{\pi_{\text{C}}}_{t+2}]](\boldsymbol{s}) \overset{(b)}{=} [T_{\dot{\text{W}}}[T_{\dot{\text{D}}}\bar{V}^{\pi_{\text{C}}}_{t+2}]](\boldsymbol{s}) = [T_{\dot{\text{W}}}\bar{Q}^{\pi_{\text{C}}}_{t+1}(.,\dot{\text{D}})](\boldsymbol{s})$$

$$\overset{(c)}{\geq} [T_{\dot{\text{W}}}\max_{\dot{\text{A}}\in\dot{\boldsymbol{B}}_3}\bar{Q}^{\pi_{\text{C}}}_{t+1}(.,\dot{\text{A}})](\boldsymbol{s}) - \sum_{t_1=t+1}^{T} [P_{\dot{\text{W}}}P_\alpha(.,t_1-t-1)\frac{1}{6}\left(\frac{1}{2}\right)^{T-t_1}](\boldsymbol{s})$$

$$\overset{(d)}{\geq} \max_{\dot{\text{A}}\in\dot{\boldsymbol{B}}_3}[T_{\dot{\text{W}}}\bar{Q}^{\pi_{\text{C}}}_{t+1}(.,\dot{\text{A}})](\boldsymbol{s}) - \sum_{t_1=t+1}^{T} \frac{1}{6}\left(\frac{1}{2}\right)^{T-t_1} [P_{\dot{\text{W}}}P_\alpha(.,t_1-t-1)](\boldsymbol{s})$$

$$\overset{(b)}{=} \max_{\dot{\text{A}}\in\dot{\boldsymbol{B}}_3}\bar{Q}^{\pi_{\text{C}}}_t(\boldsymbol{s},\dot{\text{A}}) - \sum_{t_1=t+1}^{T} \frac{1}{6}\left(\frac{1}{2}\right)^{T-t_1} [P_{\dot{\text{W}}}P_\alpha(.,t_1-t-1)](\boldsymbol{s})$$

$$\overset{(e)}{=} \max_{\dot{\text{A}}\in\dot{\boldsymbol{B}}_3}\bar{Q}^{\pi_{\text{C}}}_t(\boldsymbol{s},\dot{\text{A}}) - \sum_{t_1=t}^{T} \frac{1}{6}\left(\frac{1}{2}\right)^{T-t_1} P_\alpha(\boldsymbol{s},t_1-t)$$

where in **(a)** $\dot{\mathrm{E}}$ is any distribution in $\Delta(\dot{\mathrm{C}}, \dot{\mathrm{W}})$ and this step holds because of Lemma 5, **(b)** holds because of the exchangeability property of Lemma 6, **(c)** is true by induction and monotonicity of Bellman operator, in **(d)** the max operators change from being specific to any next state $s'$ at time $t+1$ to being just one max operator that has to choose a single optimal gain distribution in state $s$ at time $t$, **(e)** holds by definition as for any $t_2$, (here the last equality holds because $s \neq s_\alpha$) $[P_{\dot{\mathrm{W}}} P_\alpha(., t_2)](s) = \mathrm{E}_{s' \sim p(.|s, \dot{\mathrm{W}})}[P_\alpha(s', t_2)] = \mathrm{E}_{s' \sim p(.|s, \dot{\mathrm{W}})}[P(T_{\alpha, s'} = t_2)] = P_\alpha(s, t_2 + 1)$. □

**Lemma 9.** *For $t > 0$ and any $s$,*

$$P_\alpha(s, t) \leq \frac{2}{t}\sqrt{\frac{2}{\pi}}.$$

*Proof.* Using the connection between the TWIN-COMB strategy and a simple random walk in Proposition 1, a formula can be found for $P_\alpha(s, t)$ from the classical "Gambler's ruin" problem, where one wants to know the probability that the gambler reaches ruin (here state $s_\alpha$) at any time $t$ given an initial capital in dollars (here $i_s$ as defined in Proposition 1). The gambler has an equal probability to win or lose one dollar at each round and has no upper bound on his capital during the game. Using [7] (Chapter XIV, Equation 4.14) or [18] we have $P_\alpha(s, t) = \frac{i_s}{t}\binom{t}{\frac{t+i_s}{2}}2^{-t}$, where the binomial coefficient is 0 if $t$ and $i_s$ are not of the same parity. The technical Lemma 14 completes the proof. □

We now state our main result, connecting the value of the COMB adversary to the value of the game.

**Theorem 1.** *Let $K = 3$, the regret of* COMB *strategies against any learner $\bar{p}$, $\min_{\bar{p}} V^T_{\bar{p}, \pi_C}$, satisfies*

$$\min_{\bar{p}} V^T_{\bar{p}, \pi_C} \geq V_T - 12\log^2(T+1).$$

We also characterize the minimax regret of the game.

**Theorem 2.** *Let $K = 3$, for even $T$, we have that*

$$\left| V_T - \binom{T+2}{T/2+1}\frac{T/2+1}{3*2^T} \right| \leq 12\log^2(T+1), \qquad with \quad \binom{T+2}{T/2+1}\frac{T/2+1}{3*2^T} \sim \sqrt{\frac{8T}{9\pi}}.$$

In Figure 4 we introduce a COMB-based learner that is denoted by $\bar{p}_C$. Here a state is represented by a vector of 3 integers. The three arms/experts are ordered as (1) (2) (3), breaking ties arbitrarily. We connect the value of the COMB-based learner to the value of the game.

**Theorem 3.** *Let $K = 3$, the regret of* COMB-*based learner against any adversary $\pi$, $\max_\pi V^T_{\bar{p}_C, \pi}$, satisfies*

$$\max_\pi V^T_{\bar{p}_C, \pi} \leq V_T + 36\log^2(T+1).$$

$$\boxed{\begin{aligned} p_{t,(1)}(s) &= V^{\pi_C}_{t+1}(s + e_{(1)}) - V^{\pi_C}_t(s) \\ p_{t,(2)}(s) &= V^{\pi_C}_{t+1}(s + e_{(2)}) - V^{\pi_C}_t(s) \\ p_{t,(3)}(s) &= 1 - p_{t,(1)}(s) - p_{t,(2)}(s) \end{aligned}}$$

Figure 4: A COMB learner, $\bar{p}_C$

Similarly to [2] and [14], this strategy can be efficiently computed using rollouts/simulations from the COMB adversary in order to estimate the value $V^{\pi_C}_t(s)$ of $\pi_C$ in $s$ at time $t$.

# 6 Discussion and Future Work

The main objective is to generalize our new proof techniques to higher dimensions. In our case, the MDP formulation and all the results in Section 4 already holds for general $K$. Interestingly, Lemma 3 and 4 show that the COMB distribution is the balanced distribution with highest per-step regret in all the states $s$ such that $|\mathbf{x}(s)| \leq 2$, for arbitrary $K$. Then assuming an ideal exchangeability property that gives $\max_{\dot{\mathrm{A}} \in \dot{B}} \dot{\mathrm{A}}\dot{\mathrm{C}}\ldots\dot{\mathrm{C}} = \max_{\dot{\mathrm{A}} \in \dot{B}} \dot{\mathrm{C}}\dot{\mathrm{C}}\ldots\dot{\mathrm{C}}\dot{\mathrm{A}}$, a distribution would be greedy w.r.t the COMB strategy at an early round of the game if it maximizes the per-step regret at the last round of the game. The COMB policy specifically tends to visit almost exclusively states $|\mathbf{x}(s)| \leq 2$, states where COMB itself is the maximizer of the per-step regret (Lemma 3). This would give that COMB is greedy w.r.t. itself and therefore optimal. To obtain this result for larger $K$, we will need to extend the exchangeability property to higher $K$ and therefore understand how the COMB and TWIN-COMB families extend to higher dimensions. One could also borrow ideas from the link with pde approaches made in [6].

**Acknowledgements**

We gratefully acknowledge the support of the NSF through grant IIS-1619362 and of the Australian Research Council through an Australian Laureate Fellowship (FL110100281) and through the Australian Research Council Centre of Excellence for Mathematical and Statistical Frontiers (ACEMS). We would like to thank Nate Eldredge for pointing us to the results in [18] and Wouter Koolen for pointing us at [19]!

## Footnotes

[1] [19] also provides an upper-bound that is suboptimal when $K = 3$ even after optimization of its parameters.

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
