[Reviews · NeurIPS 2017]

Reviewer 1



Viewing the online prediction problem with expert advice as a repeated two-player zero-sum game between an adversary and a learner, the paper studies minimax strategies. The paper shows that the previously introduced adversary strategy "COMB" is optimal when the number of experts (i.e. K) is 3. Additionally, it also provides a corresponding COMB-based optimal learner in this setting K=3. Furthermore, the paper finds the value of game (regret) which introduces the optimal constant in the \sqrt(T) term. The paper is well-written. It clearly states the problem, the current literature, and its contributions. The paper outlines the approach to establish the results, and then, walk the reader through the theoretical results with provided proofs either in the main text or in the appendix. The paper seems technically sound and it provides sufficient theoretical analysis and proofs. From the reviewer's perspective, the paper has theoretical significance as it introduces the use of MDP, and consequently, interesting adversary representations via the introduced state space in the context of online learning. Given this framework, the paper introduces various interesting properties e.g. exchangeability. To conclude, the paper seems to contribute not only new results, but also new approaches possibly extendable to larger K's. TYPOS: Appendix -- Figures 5-8: the table contents are not fully adjusted for 2, w, and v.

Reviewer 2



Summary: This paper derives tight upper/lower matching bounds of the regret for the 3-expert problem. For the horizon T, the gap of lower and upper bound is 2log^2 T. The proof is reduced to an analysis of a Markov decision process. The algorithm achieving the bound is a variant of COMB strategy. Comments: The technical contribution of the paper is to solve an open problem of finding the optimal min-max regret bound for finite horizon T and K=3. The proof seems highly non-trivial. Although the expert problem in online learning is extensively studied and standard, the paper tries to make the theory finer. The result would be useful in the learning theory community.

Reviewer 3



In Minimax Optimal Players for the Finite-Time 3-Experts Prediction Problem. The authors study minimax prediction in the 3-expert case. The paper is carefully written and surprisingly pleasant to read for a paper presenting a very technical result. The setting corresponds to the hedge setting over K=3 arms where the learner plays and index from [K] and the adversary plays a gain vector from {0,1}^K for T rounds. Previously it had been shown that the hedge learner achieves the asymptotic minimax regret as K -> infinity. Much effort and progress has been previously made on variants of this setting in the attempt to characterize the minimax rates for closely related problems. The main technical result of the authors is to prove that a previously proposed adversary Comb achieves the minimax rate up to and an additive term as well to provide a learner that achieves the minimax rate up to an additive term. In terms of impact the result is rather technical and may not have wide appeal. As a reviewer I am only familiar with basic upper and lower bounds for MWA and not familiar with the literature on minimax results for this setup so there my review is limited in its ability to assess the quality of the technical step taken by these results. One concern I had was with the title of the paper. For the sake of the precision the title “Minimax Optimal Players for the Finite-Time 3-Experts Prediction Problem” I suggest that the title be preceded by “Near” as in fact they are not shown to be minimax optimal but up to an additive term.

Reviewer 4



The paper studies the classic prediction with experts advice problem. There are a finite number k of experts and a finite number T of rounds. There is a player that makes sequential decisions for T rounds based on the advice of the k experts, and his goal is to minimize the maximum regret he can experience (minimax regret). Naturally, the optimal adversarial strategy is a key quantity to study here. This paper takes up the conjectured minimax optimal adversarial strategy called "Comb strategy" in the Gravin et al. paper and shows that it is indeed minimax optimal in the case of 3 experts. The high level proof structure is to show that the comb strategy is optimal assuming that in all the future rounds the comb strategy will be employed and then induct. (Comb strategy is not the exact optimal though --- thus one has to lose a bit in terms of optimality, and bound how the error accumulates as one inducts) One of the useful lemmas in doing this is the "exchangeability property", which shows that deviating from comb strategy in exactly one round will give a regret that is independent of which round one deviates off of it. The need for this property arises naturally when one takes the induction proof structure above. The proof of this property proceeds by considering another adversarial strategy twin-comb, also proved to be optimal for geometric horizon setting by Gravin et al., and then showing that playing Comb + non-comb is same as playing twin-comb + comb where non-comb is a mixture of {Comb, twin-comb}. The question is how to generalize this for arbitrary number of experts k. For general k, is it possible to push this idea through with the generalized twin-comb? And what would be the appropriate generalization? This is not immediately clear from the paper, but I still think the contribution as it is already meets the NIPS bar, and certainly has some neat ideas. Given that this paper takes a conjectured optimal strategy, whose optimality for a small number of experts can be verified directly by writing a computer program to compute optimal adversarial values, its primary value can lie only in the new techniques it introduces. The main challenge is that since the comb adversary is not the exactly optimal adversary, all the nice properties of optimality are gone, and one has to control how the errors of this suboptimal strategy accumulate in order to establish approximate optimality. The "exchangeability" property is the main new idea for doing this. There is also the other nice contribution of proving that the player's algorithm that flows out of the comb adversary, which itself is not the exactly optimal but only approximately optimal, is an approximately optimal algorithm. Usually, when the adversary is not optimal, proving optimality for the algorithm that simulates it is not straight-forward. While the techniques don't immediately generalize, the general approach of controlling COMB adversary's errors with an exchangeability technique seems like the way to go for general k. Given this, and the fact that the work is technically interesting, I would say it meets the NIPS bar.